# Aneurysm-on-a-Chip: Setting Flow Parameters for Microfluidic Endothelial Cultures Based on Computational Fluid Dynamics Modeling of Intracranial Aneurysms

**DOI:** 10.3390/brainsci12050603

**Published:** 2022-05-05

**Authors:** Aisen Vivas, Julia Mikhal, Gabriela M. Ong, Anna Eigenbrodt, Andries D. van der Meer, Rene Aquarius, Bernard J. Geurts, Hieronymus D. Boogaarts

**Affiliations:** 1Applied Stem Cell Technologies, University of Twente, 7522 NB Enschede, The Netherlands; a.g.desavivas@utwente.nl (A.V.); anna.eigenbrodt@gmx.net (A.E.); andries.vandermeer@utwente.nl (A.D.v.d.M.); 2Multiscale Modeling and Simulation Group, Department of Applied Mathematics, University of Twente, 7522 NB Enschede, The Netherlands; julia.mikhal@gmail.com (J.M.); Gabriela.meilani89@gmail.com (G.M.O.); b.j.geurts@utwente.nl (B.J.G.); 3Department of Neurosurgery, Radboud University Medical Center, 6525 XZ Nijmegen, The Netherlands; rene.aquarius@radboudumc.nl

**Keywords:** intracranial aneurysm, aneurysm, organ on a chip, aneurysm on a chip, endothelial cells, computational fluid dynamics

## Abstract

Intracranial aneurysms are pouch-like extrusions from the vessels at the base of the brain which can rupture and cause a subarachnoid hemorrhage. The pathophysiological mechanism of aneurysm formation is thought to be a consequence of blood flow (hemodynamic) induced changes on the endothelium. In this study, the results of a personalized aneurysm-on-a-chip model using patient-specific flow parameters and patient-specific cells are presented. CT imaging was used to calculate CFD parameters using an immersed boundary method. A microfluidic device either cultured with human umbilical vein endothelial cells (HUVECs) or human induced pluripotent stem cell-derived endothelial cells (hiPSC-EC) was used. Both types of endothelial cells were exposed for 24 h to either 0.03 Pa or 1.5 Pa shear stress, corresponding to regions of low shear and high shear in the computational aneurysm model, respectively. As a control, both cell types were also cultured under static conditions for 24 h as a control. Both HUVEC and hiPSC-EC cultures presented as confluent monolayers with no particular cell alignment in static or low shear conditions. Under high shear conditions HUVEC elongated and aligned in the direction of the flow. HiPSC-EC exhibited reduced cell numbers, monolayer gap formation and cells with aberrant, spread-out morphology. Future research should focus on hiPSC-EC stabilization to allow personalized intracranial aneurysm models.

## 1. Introduction

Intracranial aneurysms are pouch-like extrusions from the vessels at the base of the brain which can rupture causing a subarachnoid hemorrhage (SAH) with devastating consequences. The incidence of aneurysmal SAH is about 6 per 100,000 per year and is a major cause of death and disability [1,2]. It affects relatively young people with a mean age at presentation between 50 and 60 years [3]. About 3% of the general population carry an unruptured intracranial aneurysm (UIA) [4,5]. Flow-induced wall-shear stress (WSS) acting on the endothelial surface contributes to aneurysm formation [6]. To understand the role of WSS numerical simulations of hemodynamics, so-called computational fluid dynamics (CFD), have been conducted. CFD studies recognized that wall shear stress contributes to IA formation. In fact, both low and high WSS can drive IA growth [7,8,9]. On a cellular level inflammation is identified as a key contributor to aneurysm pathogenesis [10] The formation and the progression of IA results from endothelial dysfunction, an increasing inflammatory response and phenotypic modulation of vascular smooth muscle cells into a pro-inflammatory phenotype [10]. Next, apoptosis of cellular components of the vessel wall will eventually result in aneurysmal dilatation and rupture in certain aneurysms. The endothelial dysfunction is driven by inflammatory mediators, wherein NF-kB plays an initiating and central role [11].

An intracranial aneurysm (IA) can be treated by several invasive interventions with treatment-specific and varying complication risks. Therefore, it is essential to identify patients at risk for rupture. Size and site of the aneurysm are risk factors for rupture [12]. Paradoxically most SAHs are from small aneurysms, since small aneurysms are much more common than large aneurysms [13]. Aneurysm morphology and growth are additional risk factors. Risk models have been proposed without the inclusion of all relevant factors (e.g., smoking, familial history) [12,13]. Aneurysm wall enhancement associated with inflammation, as seen on magnetic resonance (MR) imaging after gadolinium, is also associated with increased rupture risk [14,15,16,17]. Current models are not capable of a precise prediction of rupture risk. Therefore, new models to capture these processes are needed.

Other researchers have shown that there is a correlation between cellular signals in response to controlled flow dynamics in vitro and signalling in locations with similar flow dynamics in vivo [18]. These findings demonstrate the added value of developing in vitro models that would exactly capture specific the flow dynamics of aneurysms in vivo.

Roughly, there are two approaches to achieve this. The first approach relies on directly replicating aneurysm geometry in vitro [19,20,21]. However, due to their small size and abstract geometries, these models do not fully replicate the flow dynamics as found in vivo. Alternatively, in vitro models that capture flow dynamics of specific areas in an aneurysm as derived from CFD on patient aneurysms can also be established [11]. However, current examples of this approach are based on large flow systems, such as cone-and-plate and parallel plate chambers. Such large culture systems have a high consumption of medium and cells. Moreover, they are not easy to parallelize, which means they can only capture the flow dynamics of a single or a few locations of an aneurysm in a single study.

Here, we report an approach in which we first perform CFD on patient aneurysm geometries, followed by mimicry of location-specific flow dynamics in a microfluidic chip. We demonstrate that each location can be modelled in an individual microfluidic channel, thereby not having to rely on mimicking a full aneurysm in a single model. Moreover, our approach is based on microfluidics, meaning that consumption of media and cells is low, and that there is room for future parallelization and upscaling.

Our study represents the first step towards an approach in which flow dynamics of multiple areas of an in vivo aneurysm would be captured in many parallelized individual microfluidic channels. We report the first results of this patient specific aneurysm-on-a-chip model and discuss limitations and possible future research directions.

## 2. Materials and Methods

### 2.1. Computational Fluid Dynamics on CT Imaging Data

Approval of the study protocol was obtained from the local institutional review board (no. 2016-2419). A patient harboring an intracranial aneurysm was selected randomly. High-resolution CT imaging data can be segmented precisely to uncover the patient-specific targeted aneurysm. The raw data was recorded at a spatial resolution of 2 pixels per mm. At this refinement the local vasculature and the aneurysm sac can be clearly displayed. The numerical CFD simulations employ an immersed boundary (IB) method in which the pulsatile flow in the highly complex domain is represented on a Cartesian grid [22]. The spatial resolution of the CT imagery can be directly translated to this Cartesian representation, resulting in 30–40 grid cells per mm in the most refined simulations. We discretize the governing Navier–Stokes equations for incompressible flow using a second-order accurate finite volume method. The flow solver used for the simulations and post-processing of the wall shear stress is implemented in the open-source platform OpenFOAM version 2.0 [https://openfoam.com (accessed on the 1 March 2019)]. This allows large-scale parallel computations of the time-dependent flow and enables to capture all relevant dynamics that arises in the course of a heartbeat. By comparison of flow predictions at different spatial resolutions we could quantify the numerical accuracy achieved in the simulations and the corresponding WSS levels.

Clinical imaging data in DICOM format was imported into the open-source software HOROS version 3.3.5 (Horos is sponsored by Nimble Co LLC d/b/a, Annapolis, MD, USA, 2020) for conversion in STL format. Further processing and some smoothing of the recorded imagery was performed on the basis of the 3D creation suite BLENDER version 2.79. Horos is a free and open-source code software (FOSS) program that is distributed free of charge under the LGPL license at Horosproject.org and sponsored by Nimble Co LLC d/b/a Purview in Annapolis, MD USA. The resulting STL surface representation was subsequently transferred into a volumetric format to enable a volume penalization method as a basis for the detailed flow simulation. The latter step was realized using the open-source package OpenFOAM, carrying out the CFD analysis. CFD calculations were performed based on the clinical imaging (CT) and based on a typical input flow velocity of 0.2 m/s, corresponding to a Reynolds number of 250, in which the diameter of the supplying blood vessel of 4 mm was adopted as a length scale and the kinematic viscosity of blood of 3.25 × 10^−6^ m^2^/s was used.

### 2.2. Microfluidic Chip Fabrication

Microfluidic chips were fabricated based on conventional polydimethysiloxane (PDMS) soft lithography (Figure 1). A polymethylmethacrylate (PMMA) mold containing structures with a length × width × height of 56 × 1.2 × 0.1 mm, resp. was prepared by computer numerical control (CNC) milling (Datron Neo, Datron) (Appendix A). A mixture of 10:1 crosslinker-base reagent PDMS (Sylgard 184, Mavom Chemical Solutions) was poured on the molds and cured for three hours at 70 °C. The slabs of PDMS were removed from the molds and 1 mm diameter inlets were created with a biopsy punch (Robbins Instruments). For the bottom surface of the chips, glass microscope slides (22 mm × 70 mm) were spin-coated for 3 s with 500 rpm and for 60 s with 1500 rpm with the same PDMS mixture and cured for one hour at 70 °C. The PDMS surfaces were activated with an oxygen plasma sterilizer (Femto Science, Hwaseong, South Korea) for 40 s at 50 W and pressed together to achieve permanent bonding. Next, the channel surfaces were silanized by adding 3% (*v*/*v*) 3-minopropyl)triethoxysilane (APTES; Sigma Aldrich, Zwijndrecht, The Netherlands) for 2 min, after which they were washed twice with 70% ethanol and air dried. Silanization promotes protein coating and subsequent cell attachment, as described previously [23]. Afterwards, the chip was dried thoroughly by baking for 30 min at 70 °C and was sterilized under UV-light (Laminar flow cabinet, Telstar, Woerden, The Netherlands) for 30 min. Finally, the chips were coated with rat tail collagen type I (VWR) in phosphate-buffered saline (PBS; Thermo Fisher Scientific, Breda, The Netherlands) for 30 min at 37 °C and 5% CO_2_ and rinsed once with PBS. A detailed material list is provided in Appendix A.

### 2.3. Endothelial Cell Culture inside Microfluidic Chips

Human umbilical vein endothelial cells (HUVEC; Lonza, Geleen, The Netherlands) were cultured in Endothelial Growth Medium-2 (EGM-2, PromoCell, Heidelberg, Germany) supplemented with 1% (*v*/*v*) penicillin/streptomycin (Thermo Fisher Scientific, Breda, The Netherlands). Human-induced pluripotent stem cell-derived endothelial cells (hiPSC-EC) were differentiated based on a previously published protocol and cultured in endothelial cell basal serum-free growth medium (EC-SFM; Thermo Fisher Scientific, Breda, The Netherlands) containing 1% platelet-poor plasma-derived serum, 30 ng/mL vascular endothelial growth factor (VEGF, Miltenyi, Bergisch Gladbach, Germany) and 20 ng/mL basic fibroblast growth factor (bFGF, Miltenyi, Bergisch Gladbach, Germany) [24]. All cells were cultured at 37 °C and 5% CO_2_ in a humidified incubator. Cells were cultured in polystyrene T-75 flasks (Greiner, Alphen aan den Rijn, The Netherlands) coated with rat tail collagen type I in PBS at 37 °C and 5% CO_2_. Cells were grown to 80–90% confluence and all experiments were conducted using cells from passages 4–7. For splitting or seeding the cells into the channels, HUVEC were washed with PBS and trypsinized with Trypsin-EDTA (0.5% (*v*/*v*; Thermo Fisher Scientific, Breda, The Netherlands) in PBS for 3 min at 37 °C and 5% CO_2_. A volume of 25 μL containing a suspension of 3.5 × 10^6^ cells/mL was pipetted into the channels of the microfluidic chip. The chip was flipped upside down and incubated for 30 min, allowing cells to attach to the top of the channel.

### 2.4. Exposure to Shear Stress in Microfluidic Chips

For exposure of endothelial cells to shear stress, the chips were connected to a fluidic setup (Fluigent, Le Kremlin-Bicêtre, France) with a pressure control system (MFCS-EZ), a valve (L-Switch), a flow sensor (Flow-Unit L) and medium reservoirs. Apart from the pressure control system, all components were kept inside a humidified incubator at 37 °C and 5% CO_2_. All components were connected by fluorinated ethylene-propylene (FEP) tubing with an outer diameter (OD) of 1/16″ and inner diameter (ID) of 0.020″ (Fluigent, Le Kremlin-Bicêtre, France), polyetheretherketone (PEEK) tubing (ID 0.020″, OD 0.060″) and silicone tubing (ID 1 mm, OD 3 mm; Versitec VWR, Amsterdam, The Netherlands) plugged into blunt needles (G18) with Luer-lock PEEK ferrules (3/16″, Agilent Technologies, Santa Clara, CA, USA) (Figure 1). The shear rate resulting from specific volumetric flow rates was calculated using the approximation γ = 6Q/h^2^w, with γ shear rate (in s^−1^), Q volumetric flow rate (in m^3^/s) and h and w the height and width of the channel (in m), respectively [25]. Shear stress (τ; in Pa) was calculated by τ = γμ, with μ the dynamic viscosity of water at 37 °C (approximately 7 × 10^−4^ Pa·s).

Cells were subjected to shear stresses of 0.03 Pa and 1.5 Pa for 60 h by setting the corresponding volumetric flow rates at 8.67 × 10^−10^ m^3^ s^−1^ and 4.33 × 10^−9^ m^3^ s^−1^ on the setup, respectively.

### 2.5. Imaging of Endothelial Cells

After flow experiments, cells were washed with PBS and fixed in 4% formaldehyde (Thermo Fisher Scientific, Breda, The Netherlands) in PBS for 10 min at room temperature. Phase contrast images were taken with a Nikon Eclipse Ts2 controlled by NIS-Elements BR software (Nikon Europe, Amsterdam, The Netherlands).

## 3. Results

The imaging of a patient with a posterior communicating artery aneurysm was used (Figure 2). The CFD model of the pulsatile flow enables a precise assessment of the flow and forces that arise over time. Using typical values of the kinematic viscosity, the maximal local flow velocity as estimated from noninvasive transcranial Doppler ultrasound measurements and the typical scale for the diameter of the supplying blood vessel, Reynolds number *Re* = 250, is specified. This computational setting established regions of high local shear in the downstream connection of the aneurysm sac and the vessel as well as regions of low local shear (Figure 2). The simulations yield characteristic locations of low and high shear stress, i.e., the highest shear is attained near the downstream connecting rim between the aneurysm sac and the natural vessel, while the lowest shear stress is found near the centers of the sidewalls of the aneurysm in the direction normal to the mean flow. These locations are indicated by the green-colored markers (Figure 2).

High and low shear stress locations in the computed flow correspond to high and low shear stress conditions used in the cell culture experiments.

### Exposure of Endothelial Cells to Computed Shear Stress

Endothelial cells were cultured in a microfluidic chip and exposed to either 0.03 Pa or 1.5 Pa shear stress, corresponding to regions of low shear and high shear in the computational model, respectively. The cells were exposed to shear stress for 24 h, and as a control, cells were cultured under static conditions for the same time period. At the end of the experiment, the cells were imaged using phase-contrast microscopy to assess their morphology (Figure 3). Both HUVEC and hiPSC-EC cultures presented as confluent monolayers with no particular cell alignment in static or low shear conditions. HUVEC monolayers displayed a typical cobblestone morphology, whereas hiPSC-EC had a more elongated morphology, which is typical for these respective cell types. Under high shear conditions, cell type-specific changes in morphology were found. HUVEC elongated and aligned in the direction of the flow, while hiPSC-EC exhibited reduced cell numbers, monolayer gap formation and cells with an aberrant, spread-out morphology.

## 4. Discussion

In this article, we describe several steps to capture relevant aspects of intracranial aneurysm modeling on a microfluidic device in a specific patient to strive for personalized medicine. Individual flow conditions could be derived from the patient and be transferred to the model in which high and low WSS values could be used. HUVEC and hiPSC-EC differed in their response to high shear stress. The alignment of HUVEC in the direction of flow is well-documented and typical [20]. The loss of the monolayer of hiPSC-EC in response to high shear is interesting. It may be the cause of substrate coating (collagen I) which is not ideal for these cells, therefore limiting the formation of focal adhesions and strong adhesion to the substrate. Another cause may be that these cells are particularly sensitive to high shear conditions since they correspond to a developmental stage where they are not typically subjected to high shear conditions yet. The phenotype (arterial, venous, capillary) of the hiPSC-EC is not yet fully defined; alignment in the direction of the flow is not typically reported for this cell type, for example.

Several studies tried to capture the pathophysiological entity of endothelial cells subjected to aneurysmal flow conditions in an in vitro setting capable for cellular readouts (Table 1). As with our proposed model the need for an actual outpouching or 3D printed aneurysm is not needed to investigate pathological consequences of abnormal flow on the endothelial cells [11,18]. This largely facilitates the in vitro setup without the need for (costly) complex 3D constructions. In order to study flow conditions relevant to the clinical situation, values from previously published literature have been proposed by some, [19,21] whereas others advocate the use of clinical imaging from the patient itself including this report [11,18,20]. Although the different WSS values used partially overlap across different studies, high WSS values WSS seems to be higher in the individual derived values compared to literature used parameters 0.05–3 Pa and 0.08–0.8 Pa, respectively. Different WSS seem to be relevant for different stages in the process of aneurysm formation. High WSS seems to be related to the origin of aneurysms; this observation is in line with the fact that aneurysms occur at bifurcation apices or outer walls of vascular bends [26]. Aneurysm growth and rupture are related to regions of low WSS [27]. More recently, it has been pointed out that both high and low wall shear stress can drive intracranial aneurysm growth and rupture [7]. In large atherosclerotic aneurysms, low WSS and high oscillatory shear index trigger an inflammatory-cell-mediated pathway leading to growth and rupture. On the other hand, high WSS and positive WSS gradient might initiate a mural-cell-mediated pathway, possibly associated with the growth and rupture of small or secondary bleb aneurysm phenotypes [7]. Most models use HUVEC to seed the medium, which seems the most readily available cells. Additionally, lung-derived human microvascular endothelial cells (HMVEC-Ls) and human aortic endothelial cells (HAECs) have been reported to create endothelial monolayers. These endothelial monolayers have been described as ‘functional’ and no mention has been made of monolayer gaps or cells with an aberrant, spread-out morphology as in our study [19]. Aneurysm formation and risk of rupture are generally considered multifactorial processes and several genetic aberrations have been identified [28]. In order to capture these elements, we proposed the use of hiPSC. However, endothelial stability should be accomplished.

### Limitations and Future Perspectives

First, both currently used cell types are not necessarily representative of the endothelium of large arteries. Future studies should include primary arterial cells (e.g., human carotid artery endothelial cells) and should take advantage of recent insights in the directed differentiation of hiPSC towards an arterial endothelial phenotype [29,30]. Second, the present model only uses average shear in a single direction. Future experiments should include changes in the magnitude and direction of shear stress over time. Third, in our current model we can only study basic cellular aspects related to aneurysm pathophysiology, such as cell morphology, cell junctional staining, and potentially immune cell adhesion. Forth, more complex models using a cell culture model have been reported to integrate smooth muscle cells and a full extracellular matrix [31]. Such models would allow more functional studies of aneurysm formation, such as crosstalk between intimal and medial tissue, extracellular matrix structure and remodeling for example.

## 5. Conclusions

The presented PDMS microchannel with patient-specific CFD parameters captures the crucial elements of intracranial aneurysm pathophysiology. The stability and reaction of HUVEC models are more reliable than hiPSC models. Future studies should focus on maintaining stable hiPSC in order to increase patient specificity enabling personalized medicine.

## Figures and Tables

**Figure 1 brainsci-12-00603-f001:**
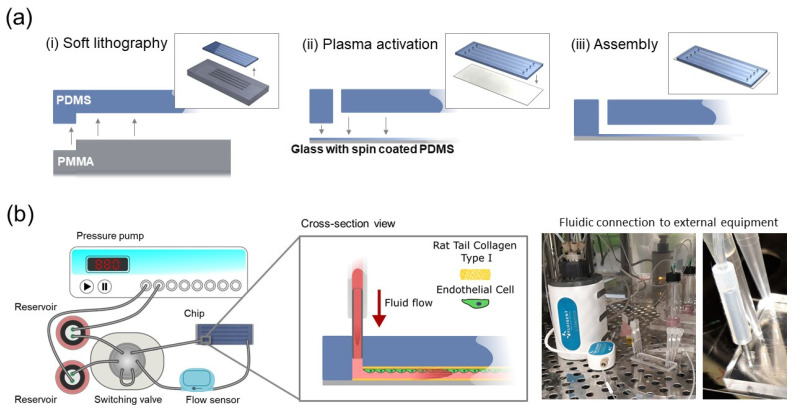
Aneurysm-on-a-chip model. (**a**) Microfluidic flow chamber was manufactured by soft lithography, by (i) pouring and crosslinking PDMS molded on mold with microstructures and spin coated on a glass coverslip, (ii) punching inlets in the PDMS slab and activating both PDMS surfaces by plasma treatment, (iii) pressing the PDMS slab on a PDMS spin coated glass slide. (**b**) The chip was connected to an integrated flow system, consisting of a pressure pump, two liquid reservoirs, a switching valve which maintains unidirectional flow in the chip and the connected flow sensor. The chip contained a monolayer of endothelial cells that were exposed to defined patterns of fluid flow. The connection to the external fluidic equipment was stablished using commercial microfluidic apparatus along with blunt needles and tube sleeves to connect the fluidic circuit to the microchannel of the microfluidic device.

**Figure 2 brainsci-12-00603-f002:**
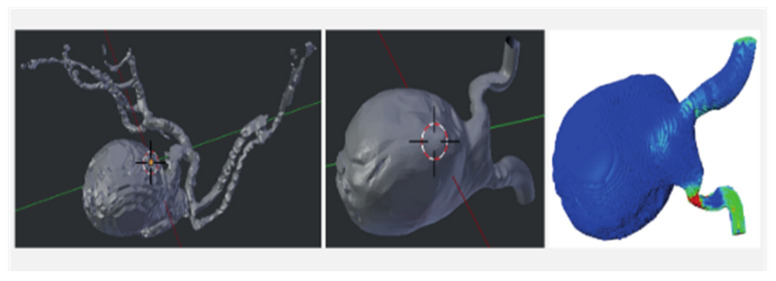
Modeling blood flow. (**left**) 3D view of an IA connected to its local vasculature. The flow in the IA is dominated by vessels directly connected to it (**middle**). A model was made of the geometry to compute the blood flow that develops in this vascular segment (**right**).

**Figure 3 brainsci-12-00603-f003:**
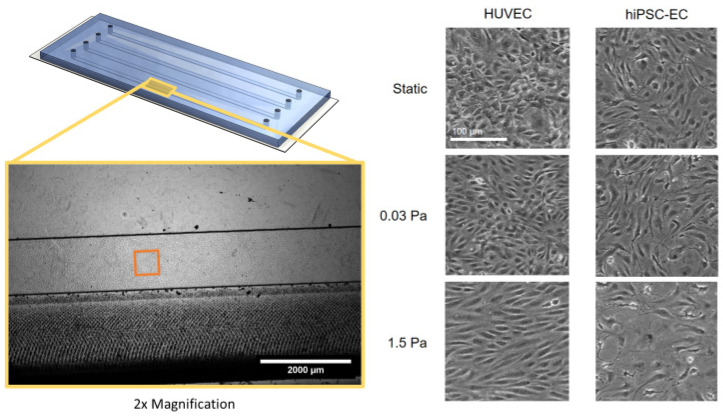
Endothelial cell morphology after 24 h of exposure to specific shear stress in a microfluidic chip channel. HUVEC (**left**) or hiPSC-EC (**right**) were cultured in microfluidic chips until confluent and were subsequently left under static conditions (**top**), treated with flow conditions causing low shear stress (**middle**) or high shear stress (**bottom**). Differences in morphology are shown by representative phase-contrast microscopy images taken after 24 h of culturing under the respective conditions. The scale bar is 100 μm.

**Table 1 brainsci-12-00603-t001:** Overview of proposed in vitro aneurysm models.

Study	Medium	Aneurysm Configuration in Medium	Flow Parameters	Cells	Primary Finding
Aoki et al. [11]	Gelatin-coated glass slides	no	CFD from CTa of 3 patients; Mixed Element Grid Generator. (0.05 or 3.0 Pa, turbulent flow)	Endothelial cells from human carotid artery	Upregulation of cell division/proliferation genes in low WSS, further augmented by turbulent flow. Increased expression of MCP-1
Baeriswyl et al. [18]	PDMS gradual and backward facing step diameter channel/pipeline on glass slide	no	CFD from 3D DSA of 4 intracranial aneurysms (2–16 dyne [0.2–1.6 Pa.])	HUVEC	Disturbed flow leads to NF-κB activation.
Kaneko et al. [20]	PDMS with fibronectin coated	Vessel replica using 3D printing	Calculated from 3d model of patient with basilary top aneurysm. (average 1.2 Pa)	Bovine carotid artery endothelial cells	Low WSS and circulating flow in apex of aneurysm. Vascular model evenly covered with monolayer EC. After 24H EC in apex irregular in shape and size, in parent artery spindle shaped and aligned with flow direction
Mannino et al. [19]	PDMS microfluidic system	Pouch in microfluidic system	From literature and CFD simulations performed to verify WSS values (~1, ~8 dyne/cm^2^ [=0.1, 0.8 Pa])	HUVEC HAEC HMVEC	Increased VCAM-1 expression correlates with low WSS
Nowicki et al. [21]	Parallel-plate flow chamber	Bifurcation and pouch in chamber	CFD from previous studies of IA (0.8 dyne/cm^2^ [0.08 = Pa])	HUVEC	Higher expression of CXCL1 and IL-8 correlates with lower WSS in aneurysm
Present report	PDMS microfluidic system	No	CFD from CTa (0.03 and 2 Pa)	HUVEC, hiPSC-EC	High WSS: HUVEC elongated and aligned in the direction of the flow. hiPSC-EC reduced cell numbers, monolayer gap formation and cells with an aberrant, spread-out morphology

CFD, computational fluid dynamics; CTa, computed tomography angiography; DSA, digital subtraction angiography; EC, endothelial cells; HAEC, human aortic endothelial cell; hiPSC, human induced pluripotent stem cell; HMVEC, lung-derived human microvascular endothelial cell; HUVEC, human umbilical endothelial cell; Pa, pascal; PDMS, polydimethylsiloxanes; WSS, wall shear stress.

## Data Availability

The data presented in this study are available on request from the corresponding author. The data are not publicly available due to privacy reasons.

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
