# Peer review of "Aneurysm-on-a-Chip: Setting Flow Parameters for Microfluidic Endothelial Cultures Based on Computational Fluid Dynamics Modeling of Intracranial Aneurysms"

_brainsci, 2022, doi:10.3390/brainsci12050603_

Round 1

Reviewer 1 Report

The work of Vivas et al., presents a microfluidic device cultured with HUVEC and human induced pluripotent stem cells-derived endothelial cells able to induce different levels of wall shear stress. The microfluidic chip developed should be able to mimic flow- induced shear stress happening during intracranial aneurysm on endothelial cells. 

According to the Reviewer's opinion, the aim of the paper is not well described as well as the results presented. It is not clear the novelty of the study, since there are in the literature already different microfluidic devices able to induce WSS. Furthermore, if the microfluidic device present in this work it's different from the ones already existing, this should be well stated and motivated.

In particular:  

1-Although the Introduction'section presents an overview of the existing microfluidic devices, a more description of the advantages and disadvantages of these devices compared with the novelty of the platform presented in the MS should be added.  

2-Then, in the methods section, it is described the immunostaining of the cells with Ve-Cadherin, DAPI and Actin green, but no images or results are presented in the MS. The authors should remove this section since it is not relevant. 

3- The results presented here show that HUVEC cells are aligned to the flow in high-shear stress values, a well-known results coming from plenty of papers already present in the literature. 

4- Discussion about the aim of the paper is missing

Accroding to these points, at this stage, the paper is not novel enough to be accepted in this journal for revision. 

Author Response

1) Although the Introduction'section presents an overview of the existing microfluidic devices, a more description of the advantages and disadvantages of these devices compared with the novelty of the platform presented in the MS should be added.

Thank you for your comment. We have better explained in the introduction section what sets our setup apart from other, previously published, devices (page 2, lines 64-84, highlighted). 

2) Then, in the methods section, it is described the immunostaining of the cells with Ve-Cadherin, DAPI and Actin green, but no images or results are presented in the MS. The authors should remove this section since it is not relevant. 

Thank you for pointing this out. We have removed this section from the manuscript (page 5, lines 176-178)

3) The results presented here show that HUVEC cells are aligned to the flow in high-shear stress values, a well-known results coming from plenty of papers already present in the literature. 

Yes, this has been demonstrated in previous papers. However, we also wanted HUVEC cells as controls when applying the same flow conditions on our setup with human induced pluripotent stem cell-derived endothelial cells

4) Discussion about the aim of the paper is missing.

The aim of the paper is to build a model which is patient specific. This has been highlighted again at the start of the discussion section (pages 6-7, lines 227-230, highlighted).

Reviewer 2 Report

The following comments may help the authors to improve the manuscript before acceptance.

  1. In the fabrication steps for the microfluidic chips. After bonding the chip, the authors performed silanization steps. Please describe in detail how this step is done and what is the purpose of using this silane layer?
  2. A list of materials (components of making the setups, etc) should be provided in the supplementary material.
  3. Details of the design of the microfluidic master mold (CAD file) should be provided.
  4. It is not clear in the paper how the authors connected the channels with the external pumping system. The connection and interfacing with external equipment should be captured and described.
  5. Figure 3 does not include the channels. It is hence not certain the cells are on or off-chip.
  6. Details of the flow rate for the perfusion cell seeding should be provided.
  7. Figure 1b does not show where the 'rat tail collagen type I' is. The author should show the 'rat tail collagen type I' layer.
  8. Figure 1a, ii, the plasma activation steps did not show the same idea as described in the text. In the text, the PDMS layer was coated on the glass slide and then the plasma treatment was followed to treat both PDMS surfaces and subsequently bonded them.

Author Response

1) In the fabrication steps for the microfluidic chips. After bonding the chip, the authors performed silanization steps. Please describe in detail how this step is done and what is the purpose of using this silane layer?

Silanization has now been described on page 3, lines 134-136.

2) A list of materials (components of making the setups, etc) should be provided in the supplementary material.

This list has been added as a supplementary file. Page 3, line 141.

3) Details of the design of the microfluidic master mold (CAD file) should be provided.

A detailed 3D file of the microfluidic master mold has been added as a supplementary file. Page 3, line 125.

4) It is not clear in the paper how the authors connected the channels with the external pumping system. The connection and interfacing with external equipment should be captured and described.

This has now been clarified in an additional panel of figure 1 (Page 4)

5) Figure 3 does not include the channels. It is hence not certain the cells are on or off-chip.

An additional panel has been added to figure 3 (Page 6).

6) Details of the flow rate for the perfusion cell seeding should be provided.

This has been stated now on page 5, lines 173-174.

7) Figure 1b does not show where the 'rat tail collagen type I' is. The author should show the 'rat tail collagen type I' layer.

Figure 1 has been adjusted and ‘rat tail collagen type I’ is now depicted.

8) Figure 1a, ii, the plasma activation steps did not show the same idea as described in the text. In the text, the PDMS layer was coated on the glass slide and then the plasma treatment was followed to treat both PDMS surfaces and subsequently bonded them.

Figure 1 has been adjusted and now shows the PDMS layer in panel ii

Round 2

Reviewer 1 Report

The authors well replied and discussed the points raised by the Reviewer. The work is now ready for publication.